# Pre-Exposure Prophylaxis (PrEP) Adherence Questionnaire: Psychometric Validation among Sexually Transmitted Infection Patients in China

**DOI:** 10.3390/ijerph182010980

**Published:** 2021-10-19

**Authors:** Xiaoyue Yu, Chen Xu, Yang Ni, Ruijie Chang, Huwen Wang, Ruijie Gong, Ying Wang, Suping Wang, Yong Cai

**Affiliations:** 1School of Public Health, Shanghai Jiao Tong University School of Medicine, Shanghai 200003, China; dd2192003@sjtu.edu.cn (X.Y.); yeschen@sjtu.edu.cn (C.X.); niy@shskin.com (Y.N.); 13916984965@yeah.net (R.C.); tanja1126@link.cuhk.edu.hk (H.W.); cynthia-dt@sjtu.edu.cn (R.G.); yingwangxun@outlook.com (Y.W.); 2Shanghai Skin Disease Hospital, Shanghai 200040, China; 3Shanghai Xuhui Center for Disease Control and Prevention, Shanghai 200030, China

**Keywords:** pre-exposure prophylaxis (PrEP), HIV prevention, questionnaire, adherence

## Abstract

Background: Ensuring adherence guarantees the efficacy of pre-exposure prophylaxis (PrEP). Methods: We conducted a cross-sectional study among 816 sexually transmitted infection (STI) patients in Shanghai. The questionnaire included self-reported demographic characteristics, self-administered items on adherence to free oral PrEP, and PrEP uptake behavior measurement. We conducted item analysis, reliability analysis, validity analysis and receiver operating characteristic (ROC) curve analysis. Results: Not all items were considered acceptable in the item analysis. The questionnaire had a McDonald’s ω coefficient of 0.847. The scale-level content validity index (CVI) was 0.938 and the item-level CVI of each item ranged from 0.750 to 1. In exploratory factor analysis, we introduced a four-factor model accounting for 79.838% of the aggregate variance, which was validated in confirmatory factor analysis. Adding PrEP adherence questionnaire scores contributed to prediction of PrEP uptake behavior (*p* < 0.001) in regression analysis. The maximum area under the ROC curve was 0.778 (95% IC: 0.739–0.817). Conclusion: The PrEP adherence questionnaire presented psychometric validation among STI patients.

## 1. Introduction

Since acquired immunodeficiency syndrome (AIDS) was first reported in 1981, it quickly became a global epidemic. The current epidemiological characteristics of the AIDS epidemic have also changed. Populations at high risk for acquiring human immunodeficiency virus (HIV) infection have shifted to sexually active groups, including people with sexually transmitted infection (STI) [1,2]. While universal antiretroviral therapy for people living with HIV is recommended and does reduce mortality, it is a treatment and not a cure [3]. Fortunately, the antiretroviral drug oral emtricitabine/tenofovir has been introduced as pre-exposure prophylaxis (PrEP), which helps reduce HIV transmission. 

Before the advent of PrEP, public health measures to prevent HIV infection, such as condom use, antiretroviral therapy, male circumcision, and regular HIV testing of high-risk populations, achieved great success in preventing HIV transmission [1]. However, the development of PrEP has brought AIDS prevention to another level, and its efficiency and safety has been proven in many clinical trials [4,5,6,7]. The initial clinical trials of PrEP have yielded different estimations of efficacy, which may be explained by varying population adherence. Two studies failed to demonstrate efficacy of PrEP, the Vaginal and Oral Interventions to Control the Epidemic (VOICE) trial with 29% population adherence, and the Preexposure Prophylaxis Trial for HIV Prevention among African Women (FEM-PrEP) trial with 37% population adherence [8,9]. Compared with other clinical trials, such as the Preexposure Prophylaxis Initiative (iPrEx) trial and the Partners Preexposure Prophylaxis (Partner PrEP) Study, where population adherence was above 50%, a low level of adherence may explain the findings in the VOICE and FEM-PrEP trials [4,10]. Nevertheless, medication adherence has an important role in the effectiveness of PrEP among people at high risk for HIV infection [11,12]. Various measures have been introduced in recent clinical studies to improve adherence, such as usage of short message service, medication level testing and electronic medicine containers with automated monitoring that remind patients to take their medication [13]. Nevertheless, unlike clinical trials in which medication adherence can be monitored and a more supportive environment can be provided by researchers, medication adherence is difficult to assess previously in the real world.

There were many studies of self-reported medication adherence, and two main approaches of adherence measurements were used at present. One approach was measurement of medication-taking behavior, which was an objective but expensive and inconvenient measure for adherence to some extent [14,15]; the other approach was identifying patient’s non-adherence, a kind of subjective measurement including barriers or beliefs related to adherence [16], such as the well-validated medication adherence scale, the Adherence Starts with Knowledge (ASK-20) scale and its brief version ASK-12, barrier items related to patient-report medication adherence and related behavior were identified [17,18]. In a previous study among transgender women sex workers, Wang explored the risk factors related to the daily use of free oral PrEP which was based on the theory of planned behavior (TPB) [19], behavior intention towards PrEP was evaluated from three aspects, including attitude (positive attitude that benefits the PrEP adherence and negative attitude), subjective norm and perceived behavioral control. Behavior intention can be directly linked to actual behavior in the framework of TPB [20]. Thus, we incorporated the TPB into the subjective adherence measurement as well as some developed adherence measurements in barriers and behaviors, in order to develop an easier, user-friendly and theoretical based tool for identifying PrEP adherence.

While many studies have assessed PrEP uptake and adherence as well as the sustainability of providing PrEP services for men who have sex with men or transgender women, patients with STI also require equivalent attention [10,19,21,22,23,24,25,26,27,28]. In a national review of the HIV infection epidemic trend in China, although the incidence of HIV infection was low among patients with STI, more HIV infections have been caused by heterosexual contact [2]. Moreover, a recent STI infection has been reported to be a risk factor for poor PrEP adherence compared with gay-identified participants [29]. Great benefits can be derived from promoting the use of PrEP where identifying medication adherence is inevitable.

Surprisingly, even though there was plenty of the literature concerning PrEP adherence, how to identify adherence in scale from reminders is less studied. In addition, considering the cost of PrEP was expensive and was known by people in the form of oral pill, we excluded the effect of PrEP cost which impeded the use of PrEP significantly, and assumed that PrEP was available for free in an oral way. Hence, what the free oral PrEP adherence questionnaire measured in our research was whether people would insist on PrEP prevention when oral PrEP is provided for free. Therefore, we chose STI patients as our targeted population, and aimed to validate a self-administrated free oral PrEP adherence questionnaire as well as to explore the predictive accuracy of the questionnaire, which would contribute to identifying patients’ adherence to PrEP in advance in a sexually transmitted infection setting. Thus, we could pay more attention to those with poorer adherence tendency and help them develop better PrEP adherence behaviors.

## 2. Methods

### 2.1. Participants

We recruited outpatients and inpatients who visited STI clinics on Wednesdays and Saturdays at two branch institutes of Shanghai Dermatology Hospital in the Jingan district (located on Qiujiang Road and Baode Road, respectively) from November 2017 to May 2018. 

The inclusion criteria were met by patients aged over 18 years old with syphilis, gonorrhea, condyloma acuminatum, and genital herpes according to the latest revised STI prevention and management measures in China. All diagnoses (including clinical and laboratory diagnoses) were conducted by doctors in the hospital.

The exclusion criteria were as follows: (1) infected with HIV (2) serious psychological or cognitive diseases; (3) unconscious status; (4) unwilling to cooperate with researchers; (5) vision or hearing loss or poor reading ability, leading to little understanding of the purpose and content of this research.

### 2.2. Data Collection

Our research team recruited all physicians working at STI clinics in two cooperative Shanghai Dermatology Hospitals. All doctors were trained on how to select patients who met the inclusion and exclusion criteria of this study. After training, doctors were responsible for informing eligible patients about the purpose and content of this research and determining their willingness to participate. The interviewer team comprised senior undergraduate or graduate students recruited from Shanghai Jiao Tong University School of Medicine, who had experience with administering questionnaire surveys. The interviewer team was responsible for conducting face-to-face interviews with patients. All interviews were conducted twice a week over nearly 7 months.

In each interview, the interviewer team invited patients one by one to a quiet room, to keep patients focused on completing the questionnaire. The interview was comprised of three phases; during the first phase, patients were provided with an informed consent form to sign. Interviewers answered all patients’ concerns in principle. During the second phase, patients were required to complete the questionnaire on their own; no private information was collected during this process. In the third phase, interviewers verified the completed questionnaire to ensure no items were missed or ignored. At the end of the interview, patients receive a cash stipend (80 RMB, nearly 12 USD) in consideration of their cooperation in this study.

### 2.3. Measurement

The questionnaire included self-reported demographic characteristics, PrEP adherence questionnaire, and PrEP uptake behavior items, to evaluate patients’ adherence to PrEP [19]. PrEP adherence questionnaire used in this research was original and self-administered, and eight items were included, based on factors found to be associated with free oral PrEP behavior in previous team work [19]. Those eight items were used to measure participants’ attitudes, subjective norms, and perceived behavioral control based on the theory of planned behavior. Behavioral intention was measured using an additional question separated from the questionnaire, to predict the actual performance of individuals. Considering the conciseness and clarity of the questionnaire items, four subscales were established: benefits, barriers, peer support, and self-efficacy. Eight items were included, with two items for each subscale. Each item on the questionnaire was scored using a three-point scale: 0 points for “disagree”, 1 point for “unclear”, and 2 points for “agree”; the total score on the questionnaire was 16 points. Higher scores indicated higher adherence to PrEP. Both positive and negative questions were included, to reduce participants’ response bias to a certain extent. (Full questionnaire was shown in Appendix A).

Participants were asked the following question apart from the adherence questionnaire: Would you be willing to take PrEP on a daily basis during the following 6 months? There were five responses, corresponding to different degrees of PrEP behavioral intention (answers: 1 = definitely not, 2 = unlikely, 3 = neutral, 4 = likely, 5 = definitely will). Given that not all behavioral intentions translate into action [30], we selected “definitely will” as the standard predicting positive behavior, in a conservative manner.

### 2.4. Statistical Analysis

For item analysis, we checked the normality via Shapiro–Wilk method, and all items did not pass the Shapiro–Wilk test. Thus, we used the critical ratio and Spearman correlation coefficient to test and evaluate the relevance and reliability of all items. In the calculation of critical ratio, participants were divided into two groups based on their total adherence scores: a high-adherence group (participants ranked among the 27% with the highest scores) and a low-adherence group (participants ranked among the 27% with the lowest scores). We conducted a *t*-test for scores of each item in the high- and low-adherence groups. A Spearman correlation test was conducted between each item score and the total score. 

For reliability analysis, McDonald’s ω coefficient and the item reliability, if item dropped, were used to evaluate the internal consistency of the questionnaire.

A panel of eight experts specializing in population health and health behavior was invited to conduct the content validity analysis, scoring the relevance of each item to the corresponding content concept, the simplicity and clarity of each item, and the ambiguity of each item. In addition, the experts gave their opinions, based on a four-point scale, in evaluating the questionnaire using the content validity index (CVI), including the item-level CVI (I-CVI) and scale-level CVI (S-CVI). 

For construct validity, factor analysis was performed. Samples were randomly divided into two groups. Group 1 (406 patients) was used for exploratory factor analysis (EFA) and Group 2 (410 patients) for confirmatory factor analysis (CFA). In the EFA, Bartlett’s test of sphericity and the Kaiser–Meyer–Olkin (KMO) value were applied for eligibility; principal component analysis was adopted to extract common factors. In the CFA, the maximum likelihood method was used to verify the fitness of the initial model. Goodness-of-fit indices such as χ^2^, degrees of freedom (df), p-value, root mean square error of approximation (RMSEA), goodness-of-fit index (GFI), adjusted goodness-of-fit index (AGFI), and comparative fit index (CFI) were adopted.

The value of average variance extracted (AVE) and construct reliability (CR) were used to evaluate convergent validity. We used the square root value of AVE and related analysis results to evaluate discriminant validity.

For predictive value and score translation, univariate logistic regression, hierarchical logistic regression, and receiver operating characteristic curve (ROC) analyses were carried out to assess the predictive value of adherence questionnaire scores in discriminating patients with positive behaviors (high-adherence group) and negative behaviors (low-adherence group), and to determine the cutoff score. 

The data were analyzed using SPSS 25.0, AMOS 22.0, R 3.6.3, and MedCalc 19.0. 

## 3. Results

### 3.1. Participant Demographic and Adherence-Related Characteristics

A total of 816 participants were recruited in this study. The mean age of participants was 38.56 years, standard deviation 13.00 years. The mean adherence score was 8.79, standard deviation 2.61. Among the 816 participants, 134 (16.42%) were aware of PrEP as a preventive measure for AIDS, and 11 patients (1.35%) had taken PrEP before the study. Table 1 shows participant demographic and adherence-related characteristics and adherence scores between the groups. The total adherence score was significantly different according to age, education level, marital status, income, sexual orientation, and knowledge about PrEP (Table 1).

### 3.2. Item Analysis

Except for item 3, all items had a critical ratio value higher than 3, with statistical significance. Except for items 3 and 4, all items had a Spearman correlation coefficient greater than 0.3, with statistical significance. Items 1, 2, and 5–8 were acceptable in the item analysis (Table 2).

### 3.3. Reliability

McDonald’s ω coefficient of total PrEP adherence for internal consistency was 0.847 and the item reliability if item dropped was reported; none of those items had a lower McDonald’s ω coefficient when they dropped, introduction of each item contributed to the improvement of reliability. (Table 3).

### 3.4. Validity

#### 3.4.1. Content Validity

The S-CVI was 0.938, and the I-CVI for each item ranged from 0.750 to 1. In general, the content validity of the questionnaire was acceptable. 

#### 3.4.2. Exploratory Factor Analysis (EFA)

The KMO value was 0.729, Bartlett’s spherical test χ^2^ = 1064.865, df = 28, *p* < 0.001, indicating that the questionnaire was suitable in the EFA. 

Four common factors with eigenvalues greater than 1 were extracted and accounted for 79.838% of the aggregate variance, which was in accordance with our theoretical four-factor structure. No item loads were less than 0.4. Whereas item 6 loaded in two subscales, it loaded heavier in factor 3 (0.728) than in factor 4 (0.455); therefore, item 6 was classified into factor 3 (Table 4).

#### 3.4.3. Confirmatory Factor Analysis (CFA) 

After EFA, a four-factor structural model was adopted for CFA, which included the following factors influencing PrEP adherence among target participants: benefits, barriers, peer support, and self-efficacy. An acceptable model fit resulted, as follows: χ^2^ = 47.1, df = 14, χ^2^/df = 3.361, RMSEA = 0.078, GFI = 0.975, AGFI = 0.936, and CFI = 0.962 (Table 5).

#### 3.4.4. Convergent and Discriminant Validity 

All subscales had a CR value and AVE value greater than 0.7 and 0.5, respectively, except for subscale support; however, the CR value of subscale support was nearly 0.7 (0.691), so the convergent validity was acceptable. The square root AVE value of each subscale was greater than the maximum correlation coefficient between the subscale and other subscales; therefore, discriminant validity was acceptable (Table 6 and Table 7).

#### 3.4.5. Predictive Value and Score Translation 

The score of the PrEP adherence questionnaire was a statistically significant predictor of PrEP uptake behavior in the initial univariate logistic regression analysis (odds ratio = 1.585, *p* < 0.001). In hierarchical logistic regression, entry of the total score improved prediction significantly (*p* < 0.001) (Table 8).

The ROC curve of the self-report PrEP adherence questionnaire score to determine participants’ actual PrEP uptake behavior and the area under the ROC curve (AUC) suggested that the PrEP adherence questionnaire performs well in distinguishing patients with high adherence and low adherence. The maximum AUC value under the curve was 0.778 (95% confidence interval: 0.739–0.817), corresponding to a cutoff score of 9 with a specificity of 72.24% and sensitivity of 74.39% (Figure 1).

## 4. Discussion

Since the introduction of PrEP as primary prevention for HIV transmission, its efficacy is evident in rigorous clinical trials [4,10]. Nevertheless, it is important to maximize the use of PrEP to best prevent HIV transmission. Adherence is one factor that has an impact on PrEP efficacy [1]. The aim of this research was to assess the reliability and validity of the free oral PrEP adherence questionnaire among patients with STI who did not have but were at high risk of HIV infection. Our results demonstrated that the PrEP adherence questionnaire had acceptable reliability and validity and could serve as a useful measurement to evaluate differences in PrEP adherence among patients with STI.

We conducted item analysis and evaluated reliability and validity in this study. In item analysis, items 3 and 4 showed disqualifying results, and should be eliminated according to the results of item analysis. However, the subscales of items 3 and 4 describe the factors that hinder adherence to PrEP, including drug side-effects and HIV stigma. The influence of these two factors on PrEP has been shown in previous studies [24,31,32], which were also wide-used items that represented barriers on adherence in some developed medication adherence scale [17,18]. In Golub’s research, PrEP stigma was found to be highly associated with HIV stigma as PrEP is designed to prevent HIV infection [32]. People tend to relate HIV stigma to concerns about being seen taking medicine by family or friends, and such concerns substantially impact adherence [25,33]. In our design of item 4, we mainly considered concerns from sexual partners, which could represent PrEP stigma to some extent. Meanwhile, we have analyzed the reasons for this situation. More specificity and clarity may be needed in the expressions of the questionnaire. Another issue that may account for this situation was that medication side-effects and HIV stigma were not the main barriers to PrEP among patients with STI. In general, considering the critical ratio value and correlation coefficient of items 3 and 4 were still statistically significant and previous research findings, we decided to reserve the room for future discussion about whether item 3 and item 4 should be eliminated. However, further investigation into barriers among the population with STI is required. 

In reliability analysis, results for internal consistency and external reliability showed that the questionnaire has acceptable reliability. In validity analysis, we first carried out EFA. According to the questionnaire design, four common factors had to be extracted during this process. Through principal component analysis, four common factors with an aggregate variance contribution rate up to 79.838% were obtained, indicating that the variance caused by the four common factors can well explain variation in the variables measured in the questionnaire. In addition, the load of each item on the subscale to which it belongs was above 0.7, demonstrating that the classification of each item to its corresponding subscale was consistent with our design purpose. Furthermore, CFA indicated that the theoretical model, divided according to the designed four subscales, could explain the actual data but more fitness was still needed to improve the value of RMSEA in order to form a better validated questionnaire structure; the above results with acceptable convergent and discriminant validity indicated that the questionnaire has good structural validity to some extent. Furthermore, the results of logistic regression and ROC curve analysis demonstrated that the predictive value of the questionnaire has enough accuracy in practice. When the adherence score obtained by patients with STI reaches 9, we believe that the patient would perform well with good adherence. The same methods for evaluating the predictive value of indicators have been used previously [34,35,36,37].

To our knowledge, no scale for PrEP adherence for STI patients was validated before. We developed the validated free oral PrEP adherence questionnaire in order to evaluate patients’ adherence in advance, so that a patient with low adherence can be detected before starting PrEP prevention, and health workers could pay more attention to helping them, and thus could yield greater benefits by improving the efficiency of PrEP application. 

## 5. Limitations

To be mentioned, the PrEP adherence questionnaire in this research was self-administered, and the reliability and validity were only evaluated in specific STI patients in this research. Before applying this PrEP adherence questionnaire to other settings, applicability needs to be considered. More data are needed in the future to prove its usefulness. Since the test–retest reliability of the questionnaire was not measured in the reliability assessment, the integrity of the reliability assessment was reduced to some extent. In addition, we could not directly observe and track participants’ medication use behavior in the real world in order to determine their adherence. An additional question was asked to assess patients’ behavioral intention as representing actual behavior; conservative prediction was adopted simultaneously to offset the uncertainty created during this process. The study population was not randomly sampled, resulting in insufficient sample representation, and the selected participants were limited to patients with STI. Owing to these limitations, whether our PrEP adherence questionnaire could be applicable to other populations and locations outside Shanghai is unknown. However, a total 816 participants were included in this research, which represents the population with STI in the survey area to some extent.

## 6. Conclusions

The PrEP adherence questionnaire is acceptable in reliability and validity. The questionnaire includes four subscales: benefits, barriers, peer support, and self-efficacy, with two items for each. The present PrEP adherence questionnaire can be used as a tool to evaluate PrEP adherence in patients with STI. 

## Figures and Tables

**Figure 1 ijerph-18-10980-f001:**
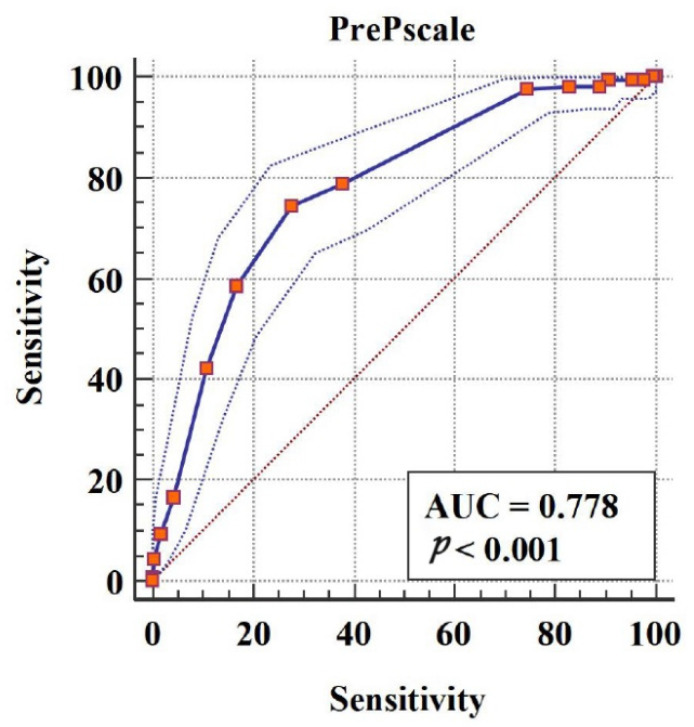
Receiver operating characteristic curve analysis of pre-exposure prophylaxis (PrEP) adherence questionnaire score for predicting PrEP uptake behavior.

**Table 1 ijerph-18-10980-t001:** Participant demographic and adherence-related characteristics.

Characteristics	*n* (%)	Adherence Score (mean ± SD)	*p*
Age			
18–30	263 (32.23%)	9.42 ± 2.56	
31–60	480 (58.82%)	8.70 ± 2.47	
61–80	73 (8.95%)	7.36 ± 2.88	<0.001
Gender			
Male	379 (46.45%)	8.73 ± 2.62	
Female	437 (53.55%)	8.84 ± 2.60	0.498
Education			
Junior high school and below	198 (24.26%)	8.44 ± 2.78	
Senior high school	187 (22.92%)	8.52 ± 2.68	
College and above	431 (52.82%)	9.15 ± 2.46	<0.001
Marital Status			
Single	232 (28.43%)	9.36 ± 2.40	
Married	520 (63.73%)	8.67 ± 2.52	
Divorced	53 (6.50%)	7.62 ± 3.49	
Widowed	11 (1.35%)	8.00 ± 2.76	<0.001
Income			
Below 3000 RMB	124 (15.20%)	8.08 ± 2.67	
3001–6000 RMB	284 (34.80%)	8.71 ± 2.81	
6001–12,000 RMB	226 (27.70%)	9.01 ± 2.39	
12001 and above RMB	182 (22.30%)	9.12 ± 2.39	0.040
Sexual Orientation			
Heterosexual	744 (91.18%)	8.78 ± 2.63	
Homosexual	25 (3.06%)	10.32 ± 2.25	
Bisexual	14 (1.72%)	9.60 ± 2.32	
Pansexual	3 (0.37%)	8.33 ± 0.58	
Unsure	30 (3.68%)	7.73 ± 2.15	0.006
Past PrEP usage history			
Yes	11 (1.35%)	8.00 ± 2.68	
No	805 (98.65%)	8.80 ± 2.61	0.350
Knowledge about PrEP			
Unknown	682 (83.58%)	8.69 ± 2.54	
Information from website and new media	78 (9.56%)	9.64 ± 2.83	
Information from healthcare workers	33 (4.04%)	8.97 ± 3.28	
Information from sexual partners, friends and other acquaintances	8 (0.98%)	8.75 ± 1.75	
Information from other sources	15 (1.84%)	8.53 ± 2.80	0.092

PrEP, pre-exposure prophylaxis.

**Table 2 ijerph-18-10980-t002:** Item analysis of the PrEP adherence questionnaire.

Item	Score	CR	*p*	Item-Total Correlation	*p*
1	2.25 ± 0.593	19.023	<0.001	0.674	<0.001
2	2.28 ± 0.630	19.245	<0.001	0.688	<0.001
3	2.07 ± 0.598	2.945	0.003	0.169	<0.001
4	2.19 ± 0.648	3.273	0.001	0.166	<0.001
5	1.98 ± 0.496	10.626	<0.001	0.492	<0.001
6	2.02 ± 0.540	16.456	<0.001	0.690	<0.001
7	2.21 ± 0.662	23.555	<0.001	0.728	<0.001
8	2.31 ± 0.682	23.609	<0.001	0.686	<0.001

PrEP, pre-exposure prophylaxis; CR, critical ratio.

**Table 3 ijerph-18-10980-t003:** Item reliability analysis of the PrEP adherence questionnaire.

Item	McDonald’s ω (If Them Dropped)
1	0.824
2	0.827
3	0.859
4	0.858
5	0.842
6	0.830
7	0.827
8	0.829
Total	0.774

PrEP, pre-exposure prophylaxis.

**Table 4 ijerph-18-10980-t004:** Exploratory factor analysis of the PrEP adherence questionnaire.

Item	Subscale 1	Subscale 2	Subscale 3	Subscale 4
1. PrEP can effectively reduce your risk of HIV infection.	0.880			
2. This medication could reduce the risk of HIV transmission to your partner.	0.893			
3. The side effects of PrEP can affect your daily life.		0.791		
4. My partner would think I do not trust him/her, if they find me taking this medication.		0.813		
5. A lot of my friend would be willing to take PrEP.			0.877	
6. My partner is supportive for me to take PrEP.			0.728	0.455
7. You are confident to use free PrEP if you want to.				0.864
8. It is up to me whether to take PrEP if PrEP is free to access.				0.853

PrEP, pre-exposure prophylaxis.

**Table 5 ijerph-18-10980-t005:** Confirmatory factor analysis of the PrEP adherence questionnaire.

χ^2^	df	χ^2^/df	RMSEA	GFI	AGFI	CFI
47.1	14	3.361	0.076	0.975	0.936	0.962

PrEP, pre-exposure prophylaxis; χ^2^: Minimum Fit Function, df, degrees of freedom; RMSEA, root mean square error of approximation; GFI, goodness-of-fit index; AGFI, adjusted goodness-of-fit index; CFI, comparative fit index.

**Table 6 ijerph-18-10980-t006:** Convergent validity analysis of PrEP adherence questionnaire.

Item	Subscale	Estimate	Standard Estimate	S.E.	CR	AVE
1	Benefit	1.000	0.861			
2	Benefit	1.024	0.831	0.054	0.834	0.716
3	Barrier	0.268	0.314			
4	Barrier	1.000	1.079	0.204	0.725	0.631
5	Peer support	1.000	0.542			
6	Peer support	1.785	0.889	0.169	0.691	0.542
7	Self-efficacy	1.085	0.820			
8	Self-efficacy	1.000	0.733	0.065	0.753	0.605

PrEP, pre-exposure prophylaxis; CR, construct reliability; S.E., standard error; AVE, average variance extracted.

**Table 7 ijerph-18-10980-t007:** Discriminant validity analysis of PrEP adherence questionnaire.

	Self-Efficacy	Peer Support	Barrier	Benefit
Self-efficacy	0.778			
Peer support	0.090	0.736		
Barrier	0.010	0.001	0.795	
Benefit	0.157	0.072	0.018	0.846

PrEP, pre-exposure prophylaxis.

**Table 8 ijerph-18-10980-t008:** Hierarchical regression analysis of PrEP adherence questionnaire score for predicting PrEP uptake behavior.

Predictive Outcome	Variable Added	Model Change	*p* Value
PrEP uptake behavior	total score	110.4	<0.001

PrEP, pre-exposure prophylaxis.

## Data Availability

All data generated or analyzed during this study are included in this published article [and its Appendix A].

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
