# Peer review of "Pre-Exposure Prophylaxis (PrEP) Adherence Questionnaire: Psychometric Validation among Sexually Transmitted Infection Patients in China"

_ijerph, 2021, doi:10.3390/ijerph182010980_

Round 1
Reviewer 1 Report
Pre-exposure Prophylaxis (PrEP) Adherence Questionnaire: planned behavioral prediction among sexually transmitted infection patients in China
Thank you for the opportunity to review this interesting article. Although it is well written and makes relevant contributions to the field, I believe it would benefit from the implementation of the following changes:
- Title. This is a psychometric validation of a questionnaire, and the title does not reflect that. I would suggest authors to replace “planned behavioral prediction” with “psychometric validation”.
- Abstract: I would suggest authors to replace “of it’s acceptable for STI patients” with “if the questionnaire presents psychometric validation…” or similar.
- Introduction: since this is a validation article, more information regarding other validated PrEP adherence questionnaires is necessary. Please provide references from other parts of the world.
- What is a free oral PrEP adherence questionnaire? Please clarify.
- Since the inclusion criteria included patients with 16 and 17 years, therefore, minor, how did authors manage the ethical problem of collecting informed consent from minors? Did parents provide authorizations? Is there a specific legal context in China that allows this?
- How many patients who were HIV+ were excluded?
- Data analysis is appropriate and well conducted.
- Authors should provide further information on how their findings can provide suggestions for future improvements in the prevention of HIV infection among the population with STI and other populations with greater risk of HIV infection
Best wishes.
Author Response
Sincere thanks for the comments for this paper. What you have mentioned in your comments will be of great help to the revision of our paper. We now have revised our manuscript according to your suggestion. Our response was included below.
- We now have changed the title according to your suggestion, which helps us make the title more accurate. Now the title in the revised version is “Pre-exposure Prophylaxis (PrEP) Adherence Questionnaire: psychometric validation among sexually transmitted infection patients in China”.
- Thank you for your advice, we now have changed the expression in Abstract according to your advice. You can find the revised expression in the revised Abstract-Conclusion.
- Thank you for your suggestion. Our questionnaire was an self-administrated questionnaire based on the results in a previous research conducted on transgender women in 2014.1 Our research is a first psychometric validation of this PrEP adherence questionnaire in STI population for the concerns that STI population were more likely to have poor PrEP adherence.2 And we now have elaborated the findings in previous research in the introduction section which was the foundation of this research. We also introduced how we designed and developed the questionnaire based on previous medication adherence measurements and PrEP adherence measurement in previous researches were described in detail in “Introduction” in the revised manuscript. (mainly in paragraph 3)
- The validation study of “Free oral PrEP adherence questionnaire” was based on the premise that PrEP was provided to STI patients for free and was delivered in oral dosage form (pill form). Because PrEP is not covered by health insurance in China, the price for a course of a PrEP treatment is expensive, so we exclude the “cost of PrEP” when we consider patients’ adherence. In addition, we used the term “oral” in order to avoid confusion, oral PrEP refers to pill form PrEP (the most familiar form for patients). Because, expect oral PrEP, there are injectable PrEP as well. We have clarified the content above and explained what the free oral PrEP adherence questionnaire measured in our research.(what a free oral PrEP adherence questionnaire measured was that whether people would insist on PrEP prevention when oral PrEP are provided for free) in the in “Introduction” in the revised manuscript. (in paragraph 5)
- Sorry for my mistake in the age inclusion criteria that I wrote it wrong before. The correct inclusion criteria about age was: aged over 18 years old. As you can find the participant’s age information in our results and table1, the age range was 18-80, no participants aged under 16 were recruited and reported, our youngest participant was 18 years old when we conducted the research. And Our research has obtained ethics approval from the Shanghai Jiao Tong University School of Medicine Public Health and Nursing Ethics Committee (approval number: SJUPN-201702), we elaborated this statement after the conclusion in “Institutional Review Board Statement”, and made corresponding revise in the “Methods-Participants” part of the manuscript.
- Sorry for the vagueness in the original manuscript. We excluded 94 patients who were HIV+, 10.3% (94/910) were HIV positive. We found this data hard to represent a HIV prevalence in STI patients, since there were quite a lot of patients declined to participant for privacy concerns.
- Sorry for the vagueness when it came to the usefulness and significance of this research in the original manuscript. We now have revised this part and explain the significance of this research in a more detailed way in the “Introduction” and at the end the “Discussion”. (revised expression in the “Introduction”: Therefore, we chose STI patients as our targeted population, and aimed to validated a self-administrated free oral PrEP adherence questionnaire as well as to explore the predictive accuracy of the questionnaire, which would contribute to identify patients’ adherence toward PrEP in advance in a sexually transmitted infection setting. Thus, we could pay more attention to those with poorer adherence tendency, and help them develops better PrEP adherence behaviors; revised expression at the end of “Discussion”: We developed the validated free oral PrEP adherence questionnaire in order to evaluate patients’ adherence in advance, that a patient with low adherence can be detected before starting PrEP prevention, and health workers could pay more attention to help them, thus could yield greater benefits by improving the efficiency of PrEP application.
1 Wang, Z. et al. Acceptability of Daily Use of Free Oral Pre-exposure Prophylaxis (PrEP) Among Transgender Women Sex Workers in Shenyang, China. AIDS Behav 21, 3287-3298, doi:10.1007/s10461-017-1869-4 (2017).
2 Jin, F. et al. Adherence to daily HIV pre-exposure prophylaxis in a large-scale implementation study in New South Wales, Australia. Aids 35, 1987-1996, doi:10.1097/qad.0000000000002970 (2021).
Reviewer 2 Report
The work presented is of enormous interest and relevance. In addition, the work is very well presented and elaborated. It is of high quality.
Below we present a series of aspects that could be presented in the manuscript:
- After the objective of the study, the authors incorporate a sentence about the usefulness of the article. It would be interesting to place this sentence elsewhere.
- The authors should include a section on the ethical aspects related to the study.
- Given that the authors include a sample of subjects under 18 years of age, how were the specific ethical aspects for this group taken into account? It is necessary to reflect carefully on this aspect given that in many countries these people cannot have access to this type of treatment.
Author Response
Sincere thanks for the comments for this paper. What you have mentioned in your comments will be of great help to the revision of our paper. We now have revised our manuscript according to your suggestion. Our response was included below.
- Thank you for your suggestion, we now have revised our abstract, and we placed our significance of research in the conclusion part of abstract.
- We have now clarified our ethical concerns in the revised manuscript in section “Institutional Review Board Statement” after Conclusion, we stated that: This research has obtained ethic approval from the Shanghai Jiao Tong University School of Medicine Public Health and Nursing Ethics Committee (approval number: SJUPN-201702).
- Sorry for my mistake in the age inclusion criteria that I wrote it wrong before. The correct inclusion criteria about age was: aged over 18 years old. As you can find the participant’s age information in our results and table1, the age range was 18-80, no participants aged under 16 were recruited and reported, our youngest participant was 18 years old when we conducted the research. And Our research has obtained ethics approval from the Shanghai Jiao Tong University School of Medicine Public Health and Nursing Ethics Committee (approval number: SJUPN-201702), we elaborated this statement after the conclusion, and made corresponding revise in the methods-participants part of the manuscript.
Reviewer 3 Report
I thank the editors for the opportunity to collaborate as a reviewer in the International Journal of Environmental Research and Public Health. I would also like to congratulate the authors of the manuscript ""Pre-exposure Prophylaxis (PrEP) Adherence Questionnaire: planned behavioral prediction among sexually transmitted infection patients in China", for the effort made in their study.
I recommend improving the following aspects:
- To use Spearman's coefficient they should have previously performed the normality and homoscedasticity tests.
- I recommend using the Omega coefficient instead of Cronbach's alpha.
- Cronbach's alpha score is low, it should be higher according to the literature (Nunnally, J. C., & Bernstein, I. H. (1994). Psychometric Theory, 3rd ed. McGraw Hill).
- According to the item-total correlation, items 3 and 4 should be eliminated (Ebel, R. L. (1965). Measuring educational achievement. Prentice Hall).
- The RMSEA should be less than 0.05 (Batista, J. M., & Coenders, G. (2012). Modelos de ecuaciones estructurales. Editorial La Muralla).
Author Response
Sincere thanks for the comments for this paper. What you have mentioned in your comments will be of great help to the revision of our paper. We now have revised our manuscript according to your suggestion. Our response was included below.
- Sorry for the vagueness in the original manuscript. We had tested the normality via Shapiro-Wilk method before, and the Shapiro-Wilk p<0.01 for all items. Because all items did not even pass the normality test first, so the Spearman's coefficient was adopted to exhibited the relationship. We now have revised the “Statistical analysis” section and clarified that why we used Spearman's coefficient for relationship test between each item score and the total score.
- Thank you for your advice, we now used the McDonald’s w as the reliability index, which was 0.847 for the questionnaire. Corresponding revises were made in the expression of “Methods-Statistical analysis”, “Results-Reliability” and “table 3”.
- I totally agreed with your suggestion. According to the item-total correlation in psychometric validation, items 3 and 4 in the questionnaire should be eliminated. We have discussed this the reason for this situation and stated that items 3 and 4 in the questionnaire should be eliminated according to the results of item-total correlation in our revised “Discussion”. Because our questionnaire was developed based on a wide-recognized theoretical model -- the theory of planned behavior (TPB) and other well-validated medication adherence such as the Adherence Starts with Knowledge (ASK-20) scale as well as the results of reliability (none of those items had a lower McDonald’s w coefficient when they dropped, introduction of each item contributed to the improvement of reliability), we decided to reserve the room for future explanation of item 3 and item 4. (Detail reasons were discussed in revised “Discussion”)
- In our results, the value of RMSEA was 0.078, which was relatively lower than reference value (0.05). This demonstrated that there is a room for improvement of our questionnaire structure. And we discussed this in our revised “Discussion”, and stated that future improvement was needed in order to form a better validated questionnaire structure.
Round 2
Reviewer 3 Report
The authors have appropriately made the suggested modifications. Congratulations on your manuscript.